# Assessment of Malnutrition in Heart Failure and Its Relationship with Clinical Problems in Brazilian Health Services

**DOI:** 10.3390/ijerph191610090

**Published:** 2022-08-15

**Authors:** Juliana Santos Barbosa, Márcia Ferreira Cândido de Souza, Jamille Oliveira Costa, Luciana Vieira Sousa Alves, Larissa Marina Santana Mendonça de Oliveira, Rebeca Rocha de Almeida, Victor Batista Oliveira, Larissa Monteiro Costa Pereira, Raysa Manuelle Santos Rocha, Ingrid Maria Novais Barros de Carvalho Costa, Diva Aliete dos Santos Vieira, Leonardo Baumworcel, Marcos Antonio Almeida-Santos, Joselina Luzia Menezes Oliveira, Eduardo Borba Neves, Alfonso López Díaz-de-Durana, María Merino-Fernández, Felipe J. Aidar, Antônio Carlos Sobral Sousa

**Affiliations:** 1Graduate Program in Health Sciences, Federal University of Sergipe (UFS), Aracaju 49060-676, Brazil; 2Graduate Program Professional in Management and Technological Innovation in Health, Federal University of Sergipe, Aracaju 49100-000, Brazil; 3Food Technology Department, São Cristovão Campus, Federal Institute of Sergipe, São Cristovão 49100-000, Brazil; 4Department of Nutrition, Campus Prof. Antônio Garcia Filho, Federal University of Sergipe (UFS), Lagarto 49400-000, Brazil; 5Clinic and Hospital São Lucas/Division, Rede D’Or São Luiz, Aracaju 49060-676, Brazil; 6Graduate Program in Health and Environment, Tiradentes University (UNIT), Aracaju 49032-490, Brazil; 7Department of Medicine, Federal University of Sergipe (UFS), São Cristovão 49100-000, Brazil; 8Division of Cardiology, University Hospital of Federal University of Sergipe (UFS), São Cristovão 49100-000, Brazil; 9Graduate Program in Biomedical Engineering, Federal Technological University of Paraná (UTFPR), Curitiba 80230-901, Brazil; 10Sports Department, Physical Activity and Sports Faculty—INEF, Universidad Politécnica de Madrid, 28040 Madrid, Spain; 11Faculty of Health Sciences, Universidad Francisco de Vitoria (UFV), 28223 Madrid, Spain; 12Group of Studies and Research in Performance, Sport, Health and Paralympic Sports—GEPEPS, Federal University of Sergipe (UFS), São Cristovão 49100-000, Brazil; 13Graduate Program in Physical Education, Federal University of Sergipe (UFS), São Cristovão 49100-000, Brazil; 14Graduate Program in Physiological Science, Federal University of Sergipe (UFS), São Cristovão 49100-000, Brazil

**Keywords:** malnutrition, cardiac insufficiency, health services

## Abstract

Malnutrition in heart failure (HF) is frequent and associated with a worse prognosis. Due to differences in investment and the profile of those assisted, the objective of this study was to evaluate the frequency of malnutrition in hospitalized patients with HF and its association with clinical outcomes in the public and private health systems. Methodology: A cross-sectional study, with 247 volunteers hospitalized with HF in three public hospitals and one private hospital in Aracaju, SE, Brazil. A subjective global nutritional assessment (SGA) and mini nutritional assessment (MNA) were performed. Results: Sample with 72.5% users of the public health system and 75.3% with malnutrition (public = 74.9%; private = 76.5%; *p* = 0.793). Regardless of the healthcare system, hospital stay (>14 days) was longer (*p* = 0.020) among those with malnutrition (48.4%) than well-nourished patients (29.5%). Malnutrition in the public system had higher mortality (7.5%; 5.8%; *p* < 0.001) and hospital transfer rate (21.1%; 0.0%; *p* < 0.001) than those in the private system. Death after discharge was observed only in the public system (*p* = 0.039). Conclusion: Malnutrition was frequent in both systems and was associated with longer hospital stays and, in the public hospital, in-hospital death and transfers.

## 1. Introduction

Heart failure (HF) is a major public health problem that affects about 26 million people worldwide [1,2]. Despite advances in the treatment of HF, malnutrition is still a frequent and neglected condition, which affects between 30 and 70% of patients with the disease [1,3,4]. This association occurs in a multifactorial and complex way, which can be a consequence of inadequate intake and absorption of nutrients or even of the systemic response to a decrease in cardiac function, which causes alterations in both anabolic and catabolic metabolism, from neurohormonal and inflammatory activation. The therapeutic approaches conventionally adopted, such as polypharmacy and restrictive diet, in addition to the clinical picture of HF, favor malnutrition, either by reducing ingestion and/or nutrient absorption [1].

It is also worth mentioning that the worsening of HF promoted by malnutrition can cause an increase in its morbidity and mortality rate and the length of hospital stay [1,4,5]. This finding can explain what has been observed in recent decades, especially in Brazil, that besides the reduction in the number of hospitalizations for HF, there has been an increase in the length of hospital stay (12.3%) and mortality rate (30.1%) due to the disease [6]. Thus, nutritional care has a relevant role in the prognosis of patients and reducing the costs associated with HF. In the United States alone, a country that does not have a universal health system like in Brazil, the cost of handling HF will reach USD 69.7 billion by 2030 [7].

In Brazil, health systems have differences in terms of investment, access, and even in the profile of the population [8]. While the public health system offers care to the entire Brazilian population, which is more than 214 million people, only 28.5% of inhabitants also have access to the private health system [8,9]. Even so, the private system is responsible for higher expenses than the public one under the justification of being more efficient. This argument is also used to justify public policies and tax incentives that favor the private health system [10,11].

In this context, questions arise about whether the differences between the types of care (public vs. private) have repercussions on the clinical picture of hospitalized patients [12,13,14,15,16]: Is the prevalence of malnutrition higher among patients in the public system? Are malnutrition patients treated in the public system more susceptible to unfavorable outcomes than patients hospitalized in the private system? Despite the relevance of the theme, there is still a lack of studies comparing the occurrence of malnutrition among patients with HF hospitalized in both types of healthcare system. Therefore, recognizing the possible presence of malnutrition and its associated factors can improve the prognosis and quality of life of patients with HF. In addition to contributing to planning more efficient and effective public policies to face the disease, the present study aims to identify the frequency of malnutrition and its association with clinical outcomes of patients hospitalized for HF in public and private health systems.

## 2. Methodology

### 2.1. Study Design

This cross-sectional, observational, analytical, census-based study used data from the VICTIM-CHF (Congestive Heart Failure) Registry. The investigation was carried out in four cardiology referral hospitals in the city of Aracaju, SE, Brazil, three of which were intended for public care (Surgery Hospital, University Hospital of the Federal University of Sergipe—HU/UFS, and Sergipe Emergency Governor João Alves Filho—HUSE, Aracaju, SE, Brazil) and one for private assistance (São Lucas Hospital—D’or Sao Luiz, Aracaju, SE, Brazil), following the steps described in Figure 1.

### 2.2. Study Sample Selection

Patients of both sexes aged 18 years or older who had a diagnosis of HF confirmed by the Framingham criteria were included. Those who had their HF diagnosis altered during hospitalization, those who were not evaluated by the nutritional screening tools, those with neurocognitive alterations that compromised verbalization, those who were in isolation or were diagnosed with COVID-19, pregnant women and lactating women were excluded from this study.

The sample was chosen by the criterion of convenience, with 407 patients being initially selected; after applying the exclusion criteria, 247 volunteers remained and attended from April 2018 to January 2021.

Data collection and completion of the Case Report Form (CRF) specific to the investigation were performed by trained researchers based on medical records (clinical data and hospital outcome), interviews (socioeconomic status, health history, lifestyle, functional classification—NYHA), and nutritional assessment (nutritional screenings and anthropometry) at the hospital bed, and finalized through a telephone call to the patient or caregiver 30 days after hospital discharge, following the Strengthening the Reporting of Observational Studies in Epidemiology (STROBE) (Figure 1) [17,18].

The study was approved by the Research Ethics Committee of the Federal University of Sergipe (Opinion number: 2,670,347). The study protocol was clearly explained and was only initiated if there was voluntary agreement by the patient after signing the informed consent form.

### 2.3. Socioeconomic, Clinical and Lifestyle Data

Socioeconomic data (gender, age, education, type of care, and per capita income) were collected. Per capita income was calculated from the ratio between family income and the number of people in the household and was presented according to the Brazilian minimum wage in force in 2022 (BRL 1212.00). Schooling was assessed from the time of the study and presented in years. In cases where the patient only reported the school level, it was considered the time that is expected to complete each stage of education adopted in Brazil, according to the Law of Directives and Bases of Education (LDB) [19].

From the interview, information was obtained about comorbidities presence (arrhythmia, depression, diabetes, dyslipidemia, peripheral arterial disease, chronic obstructive pulmonary disease, systemic arterial hypertension, and renal failure) and pathological antecedents (stroke or ischemic attack, cancer, and acute myocardial infarction). The family history of coronary artery disease was also observed.

From the medical records, data about HF diagnosis at admission and verification of whether it was maintained until discharge were collected. Additionally, the etiology, left ventricular ejection fraction (echocardiogram), admission and discharge date, and reason for leaving the hospital (discharge, transfer, or death) were observed. The admission and end of hospitalization dates were collected to obtain the length of stay.

Lifestyle was evaluated by the variables: consumption of alcoholic beverages, smoking, and level of physical activity. Habitual consumption of alcoholic beverages and cigarettes (smoking) was evaluated based on their respective dichotomous variables (yes or no). To assess the level of physical activity, the short version of the International Physical Activity Questionnaire (IPAQ) was used, classifying it as low, moderate, and high. The physical activity level classification is based on the metabolic equivalent classification (MET—minute/week) [20]. This measure expresses the energy spent to perform activities in relationto to the resting metabolic rate in a one-week period and was obtained from the inclusion of the frequency and time spent and was calculated automatically from an Excel^®^ file made available by the IPAQ [21].

HF was classified according to the left ventricular ejection fraction (LVEF) and symptom severity. The classification of insufficiency based on the ejection fraction is essential because it is associated with the presence of the main etiologies, comorbidities, and even the therapeutic response. They were classified as HFrEF—HF with reduced ejection fraction LVEF < 40%; HFmrEF—HF with mildly reduced ejection fraction LVEF between 40 and 49%; and HFpEF—HF with preserved ejection fraction, LVEF ≥ 50% [22]. This measurement was obtained from the echocardiogram, calculated by the methods of Teicholz and Simpson.

Classification based on the severity of symptoms occurs through the functional classification of the New York Heart Association (NYHA). This designation is categorized into four classes (I, II, III, and IV), defined based on the patient’s exercise tolerance, which varies from symptom absence to symptoms even at rest. Considering that NYHA III and IV are associated with worse clinical status, longer hospital stays, and risk of mortality, these were combined into a single category and, consequently, NYHA I and II formed another group [22,23].

### 2.4. Nutritional Assessment

Nutritional assessment was performed using body mass index (BMI) assessment and nutritional screening tools. Subjective nutritional assessment in all hospitals was carried out by a single team of trained researchers composed of nutrition students and nutritionists. This evaluation was performed in the hospital bed in the first days of hospitalization to minimize sample loss, due to the possibility of the patient being discharged.

The subjective global assessment (SGA) was used only in adults (≥18 to <60 years) as it is considered the gold standard for screening for malnutrition in this population [24,25]. SGA in patients with HF has a specificity of 99% and a sensitivity of 56% [25]. This tool assesses two criteria: (a) history and clinical condition (weight, food intake, gastrointestinal symptoms, functional capacity, diseases, and metabolic demand); (b) physical examination (subcutaneous fat and muscle loss, ankle and sacral edema, and ascites) [26].

The mini nutritional assessment (MNA) was used only in the elderly (≥60 years). In patients with HF, MNA has a specificity of 99% and a sensitivity of 59% [25]. This tool is divided into (a) screening (change in food consumption, involuntary weight loss, mobility, psychological stress or acute illness, neuropsychological problems, BMI) and (b) global assessment (lives alone, use of more than three medications, skin lesions or bedsores, number of meals in the day, food consumption, health self-assessment). The nutritional status assessment was performed from the sum of the screening score and global assessment [27].

Based on the evaluation of the SGA criteria and the MNA score, a new classification was created to unite the two groups of patients (adults and elderly) and unify it. Thus, all patients deemed well-nourished by the SGA and those classified by the MNA as normal nutritional status were classified as well-nourished. Patients classified by the SGA as at-risk or severely malnourished and those classified by the MNA as at-risk or malnutrition were categorized as malnourished.

To calculate BMI, weight and height were measured. All measurements were performed by the previously trained nutrition team. The average value of the three measurements, performed in sequence, was considered to minimize errors. Weight was measured using an electronic digital scale, with a maximum capacity of 180 kg and an approximation of 100 g (Seca^®^, Hamburg, Germany). Additionally, height was measured using a stadiometer (Seca^®^, Hamburg, Germany), with a usage range of 20–205 cm and markings in millimeters. BMI was classified according to the cutoff points adopted by the Ministry of Health (Low weight: BMI < 18.5 Kg/m² and ≤22 Kg/m²; Eutrophic: BMI ≥ 18.5 and <25 Kg/m² and >22 and ≤27 Kg/m²; Overweight and obesity: ≥25 Kg/m² and ≥27 Kg/m² for adults and the elderly, respectively) [28,29,30].

In the presence of edema, BMI was calculated after discounting the weight of the edema from the measured weight, estimated according to its severity, considering the following values:ankle edema (−1 kg), knee edema (−3 kg), anasarca (−10 kg), mild ascites (−2.2 kg), moderate ascites (−6 kg), severe ascites (−14 kg), mild peripheral edema (−1 kg), moderate peripheral edema (−5 kg) and severe peripheral edema (−10 kg) [31].

In cases of bedridden patients, weight and height measurements were estimated. Weight was measured from estimation equations proposed by Chumlea et al. (1988) and height from estimation equations proposed by Chumlea, Guo, and Steinbaugh (1994) [32]. To use the estimation equations, it was necessary to measure knee height (AJ) and arm circumference (AC); measurements were performed with a flexible and inelastic tape measure, subdivided into millimeters. Knee height measurement was performed with the patient in the supine position, with the ankle and right knee flexed at a right angle. The inelastic tape was positioned on the bottom of the feet, in the middle of the heel, up to the head above the femoral condyles, and close to the kneecap. The measurement of arm circumference was performed at the midpoint between the acromion and the olecranon. As well as knee height, this measurement was also performed with the patient in the supine position [33].

### 2.5. Statistical Analysis

Were considered the following as outcomes: in-hospital death, hospitalization period, readmission, or emergency care up to the 30th day after discharge and death after discharge (up to the 30th day). For analysis purposes, transfer, as well as discharge or death, were considered the end of hospitalization.

Because we used data from VICTIM-CHF, a census study, sample size estimation was not performed. Numerical variables (age, years of education, per capita income, and length of stay) were categorized and presented by absolute and relative frequency. Comparison of these variables between the types of care (public vs. private), nutritional status (malnutrition vs. nourished), and these two together (malnutrition–public vs. malnutrition–private) were performed using Pearson’s chi-square test (χ²) or Fisher’s exact test, according to the data distribution.

The pattern of distribution of continuous variables was evaluated by the Kolmogorov–Smirnov test. Continuous variables were presented as mean and standard deviation, except for the length of stay, which was presented as the median and interquartile range.

Data were stored in Microsoft^®^ Excel^®^ 2019 software (Microsoft, Redmond, WA, USA), and the statistical analysis was performed in R v.4.1.1 software [34] (R Foundation, Vienna, Austria). A confidence interval of 95% (*p* < 0.05) was considered for all statistical analyses.

## 3. Results

Among the 247 patients evaluated, 50.2% were male, 54.3% elderly, 83.6% lived with a per capita income of up to 1 minimum wage (BRL 1212.00), and 78.1% had low or no schooling. The public health system had higher frequencies of adults (*p* = 0.001), illiteracy (*p* = 0.001), and lower per capita income (*p* = 0.001), opposite to what was observed in the private health system (Table 1).

In the sample, 41.2% had HFpEF, 90.7% were classified as NYHA III and IV, and the most frequent etiology was hypertensive heart disease (27.9%) (Table 1).

The health and lifestyle analysis profile identified that 28.5% of the patients had a family history of coronary artery disease, 12.6% consumed alcohol, 2.8% smoked, and 83.9% had a low level of physical activity. Among patients in the private system, a higher prevalence of a history of cancer (*p* < 0.001), coronary artery disease family history (*p* = 0.036), and a higher level of physical activity (*p* = 0.013) were observed. All smokers in the investigation were users of the public system (Table 1).

The BMI assessment allowed malnutrition identification in 16.3% of the sample, more frequent in the elderly (*p* = 0.026) (Table 2). The SGA/MNA tools, on the other hand, made it possible to identify malnutrition in 75.3% of those investigated. However, regardless of the criterion used, there was no association between malnutrition and assistance plans (*p* = 0.119), as shown in Table 2.

The length of hospital stay was high among all patients with HF. Only 23.9% were discharged in the first week of hospitalization (Table 3). The median length of stay was 13 days (IQR = 8–22). There was a higher frequency of in-hospital death (7.3%), transfer (21.9%), death after hospitalization (11.7%), and readmission or emergency visits (23.4%) in the public system than in the private system (Table 3).

Almost half (48.4%) of the volunteers remained hospitalized for more than 14 days and the occurrence of malnutrition was associated with the length of hospital stay (*p* = 0.020), regardless of the type of health system (Table 4).

Among malnutrition volunteers in the public system, 7.5% died during hospitalization and 21.1% were transferred (*p* < 0.001). No significant difference was observed between these patients during the period of hospitalization or with the outcomes after discharge (Table 5).

## 4. Discussion

The present study aimed to assess the frequency of malnutrition in hospitalized patients with HF and its association with clinical outcomes in public and private health systems. Thus, the main findings were (a) high frequency of malnutrition in hospitalized volunteers with HF; (b) an association between malnutrition and longer hospital stay, regardless of the type of healthcare service users; (c) higher mortality and hospital transfer rates among malnutrition people in the public system compared to those in the private system; (d) mortality after discharge (up to 30 days) was observed only among those assisted by the public health system.

Care to maintain adequate weight and diet is a typical guideline for cardiovascular disease prevention, but in HF, this care requires greater attention, as nutritional problems, especially malnutrition, have a direct impact on the patient’s prognosis [1,22]. Therefore, the multidisciplinary approach is considered the “gold” standard for monitoring this disease and its comorbidities [22].

In a study carried out in the United Kingdom to evaluate the agreement of six nutritional screening tools, malnutrition was identified in 6% to 60% of patients with HF according to the type of tool used for diagnosis, a lower frequency than that observed in the present investigation. Although the frequency of malnutrition in patients with HF may also varies depending on the type of patient (outpatient or hospital) and the severity of the disease [25,35] in the present study, no significant differences were observed in the frequency of risk or malnutrition, between types of assistance (public vs. private).

The use of BMI to assess nutritional status in patients with HF is not appropriate because weight may be altered by the presence of edema [6]. Although weight control is a typical recommendation in guidelines for HF, they do not mention the bias in the use of BMI in these patients nor emphasize the importance of recovering the nutritional status of malnutrition patients, as in the Brazilian and European guidelines [22,36,37]. The latest American guidelines emphasized the importance of evidence-based nutritional counseling, including the increased risk of malnutrition in patients with HF [38]. However, the one that most emphasizes the recovery of nutritional status is the 2018 Japanese Society of HF scientific statement on nutritional assessment and management in patients with HF, which was prepared to summarize current knowledge and stimulate its development [39].

When it comes to SGA and MNA for the diagnosis of malnutrition in patients with HF, they have a specificity of 99% and a sensitivity of 56% and 59%, respectively [25]. Thus, these tools are more accurate when malnutrition is already installed than risk or early-stage malnutrition, since volunteers with malnutrition may have developed this condition before hospitalization, indicating a deficiency in the therapeutic approach adopted, and reinforcing that malnutrition is a neglected problem in healthcare networks.

The presence of malnutrition was associated with longer hospital stays, regardless of the type of health service users, and was also associated with in-hospital mortality in malnutrition patients in the public system compared to those in the private health system. A similar result was observed in a multicenter cohort study carried out with 800 patients from Colombian hospitals (HF, acute myocardial infarction, pneumonia, or chronic obstructive pulmonary disease), in which the presence of malnutrition increased the length of hospital stay (1.43 ± 0.61 days), the odds of in-hospital mortality (odds ratio = 2.39), the overall mortality (odds ratio = 2.52) and increased the mean cost of hospitalization by 30.13% [40].

The average length of hospital stay for malnutrition patients was higher than that observed in a cohort study carried out with 600 patients hospitalized for HF, which had a mean length of stay of 8.6 ± 7 days [41]. It is important to emphasize that both patients in the public and private systems have the longest hospital stay associated with malnutrition, reinforcing the malnutrition investigation and treatment importance in patients with HF in the prognosis of patients of all types of assistance.

Another outcome evaluated was in-hospital mortality. The in-hospital mortality rate was high, even with advances in HF therapy observed in recent decades. Differences in mortality rates between patients in the public and private systems may be associated with the existing difference in the socioeconomic profile, such as what happens with socioeconomic development which has a strong correlation with mortality (r = 0.73; *p* = 0.00001) [42].

Finally, outcomes were evaluated at 30 days (after discharge). There was no significant difference as a function of nutritional status, only as a function of the type of care. Therefore, the difference in the socioeconomic profile may be associated not only with in-hospital death but also with death after discharge. This result was different from that observed by Sze et al. (2017 and 2018), who identified an association between malnutrition and mortality [35,43].

Compared to the private system, the public system held all death cases after discharge (up to 30 days). The mortality rate among volunteers in the public system was higher than the rate observed in South America (9%), China, and the Middle East (7%) [44]. In a comparative study between public and private systems carried out in Sergipe with patients with acute myocardial infarction, a difference in access to health services and a greater chance of mortality were observed among patients in the public system (odds ratio = 2.96; 95% CI = 1.15–7.61; *p* = 0.02) [16].

As limitations of this work, it is worth mentioning the use of a non-randomized sample, considering that the study was designed to investigate the characteristics of all patients with heart failure hospitalized in Aracaju, Sergipe, Brazil. Although this place has reference cardiology centers, it is the smallest Brazilian state, a fact that was reflected in the sample size. Another limitation concerns the follow-up time after hospital discharge and the difficulty of a more detailed anthropometric assessment in hospitalized volunteers—that is, volunteers with acute or chronic HF due to the frequent presence of edema.

This study’s results help to reinforce that malnutrition among hospitalized patients cannot continue to be neglected, regardless of whether the patient is treated by the public or private system. The use of simple and low-cost tools, such as SGA and MNA, which are more sensitive than BMI, help to understand the real scenario of the nutritional status of hospitalized patients, essential for the development of more efficient and effective treatments and the elaboration of public policies.

It is important to emphasize that despite the advances in treatment obtained in recent decades, HF is still a serious public health problem, constituting the main cause of hospitalization in the West and a considerable part of the expenses of public health services. Thus, care for malnutrition, both as a prophylactic and therapeutic measure, aims to prevent the progression of heart failure, improve prognosis, reduce the number of hospitalizations and, consequently, minimize the financial impact of this disease on public health expenditure and in the patient’s quality of life.

## 5. Conclusions

The frequency of malnutrition among hospitalized patients with HF was high, regardless of the type of care, despite the differences between the socioeconomic, clinical, and lifestyle profiles of these patients. Malnutrition was associated with a longer stay in public and private health systems and a higher rate of in-hospital mortality and transfers among malnutrition people in the public system. Outcomes after discharge were associated with the type of care and were more prevalent among patients treated in the public system.

## Figures and Tables

**Figure 1 ijerph-19-10090-f001:**
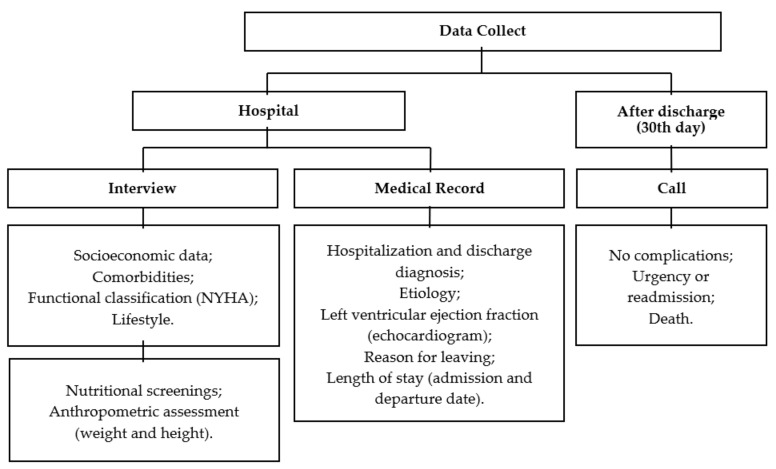
Description of data collection steps.

**Table 1 ijerph-19-10090-t001:** Socioeconomic and health history characterization of hospitalized patients with heart failure, in the public and private health systems of Brazil, from April 2018 to January 2021.

Variables	All	Private	Public	*p* ^1^
	*n* (%)	*n* (%)	*n* (%)	
**Sex**	247	68	179	0.371
Male	124 (50.2)	32 (47.1)	92 (51.4)	
Female	123 (49.8)	36 (52.9)	87 (48.6)	
**Age**	247 (100)	68 (100)	179 (100)	<0.001
<60 years old	113 (45.7)	6 (8.8)	107 (59.8)	
≥60 years old	134 (54.3)	62 (91.2)	72 (40.2)	
**Schooling (years)** ^(a)^	237	60	177	<0.001
0	52 (21.9)	5 (8.3)	47 (26.6)	
1 to 9	139 (58.6)	25 (41.7)	114 (64.4)	
≥10	46 (19.4)	30 (50)	16 (9.0)	
**Per capita income** ^2^	213	43	170	<0.001
≤BRL 1212.00/month	178 (83.6)	16 (37.2)	162 (95.3)	
>BRL 1212.00/month	35 (16.4)	27 (62.8)	8 (4.7)	
**Etiology**	247	68	179	0.028
Not Specified	22 (8.9)	12 (17.6)	10 (5.6)	
Hypertensive	69 (27.9)	20 (29.4)	49 (27.4)	
Valvulopathy	61 (24.7)	10 (14.7)	51 (28.5)	
Cardiomyopathy	45 (18.2)	13 (19.1)	32 (17.9)	
Ischemic	43 (17.4)	12 (17.6)	31 (17.3)	
Other ^3^	7 (2.8)	1 (1.5)	6 (3.4)	
**NYHA** ^4 (b)^	246	68	178	0.074
I and II	23 (9.3)	10 (14.7)	13 (7.3)	
III and IV	223 (90.7)	58 (85.3)	165 (92.7)	
**Type of heart failure** (LVEF) ^5^	213	43	170	0.592
Reduced	82 (38.9)	22 (34.9)	60 (40.5)	
Intermediary	42 (19.9)	15 (23.8)	27 (18.2)	
Preserved	87 (41.2)	26 (41.3)	61 (41.2)	
**Comorbidities**				
Arrhythmia	112 (46.1)	37 (55.2)	75 (42.6)	0.078
Depression	34 (13.8)	13 (19.1)	21 (11.7)	0.132
Diabetes	92 (37.2)	29 (42.6)	63 (35.2)	0.279
Dyslipidemia	108 (43.9)	39 (57.4)	69 (38.8)	0.009
Peripheral arterial disease	29 (11.7)	8 (11.8)	21 (11.7)	0.994
Chronic obstructive pulmonary disease (COPD)	29 (11.7)	14 (20.6)	15 (8.4)	0.008
Arterial hypertension	170 (68.8)	49 (72.1)	121 (67.6)	0.499
Renal insufficiency	39 (15.9)	12 (17.6)	27 (15.2)	0.634
**Health history (prior to hospitalization)**				
Stroke/transient ischemic attack	37 (15.0)	10 (14.7)	27 (15.1)	0.941
Cancer	17 (6.9)	12 (17.6)	5 (2.8)	<0.001
Acute myocardial infarction	57 (23.1)	19 (27.9)	38 (21.2)	0.263
Coronary artery disease (family history)	70 (28.5)	26 (38.2)	44 (24.7)	0.036
**Lifestyle (prior to hospitalization)**				
Alcoholic beverage (consumption) ^(c)^	31 (12.6)	8 (11.9)	23 (12.8)	0.848
smoking	7 (2.8)	0 (0)	7 (3.9)	0.195
Physical activity level ^(d)^	224	61	163	0.013
Short	188 (83.9)	44 (72.1)	144 (88.3)	
Intermediary	26 (11.6)	12 (19.7)	14 (8.6)	
High	10 (4.5)	5 (8.2)	5 (3.1)	

Sample size: (a) *n* = 213 (private = 43 and public = 170) for years of study. (b) *n* = 246 (private = 68 and public = 178) for NYHA, comorbidities (diabetes, dyslipidemia, and renal failure), health history (family history of coronary artery disease). (c) *n* = 246 (private = 67 and public = 179) for lifestyle (alcohol consumption). (d) *n* = 224 (private = 61 and public = 163) for lifestyle (physical activity level); Legend: ^1^ Pearson’s chi-square and Fisher’s exact tests were performed (*p* < 0.05); ^2^ Brazilian minimum wage in force in 2022. ^3^ Other etiologies: myocarditis, cor pulmonale and pericardial diseases; ^4^ New York Heart Association functional classification; ^5^ Classification of heart failure according to left ventricular ejection fraction.

**Table 2 ijerph-19-10090-t002:** Comparison of the frequency of malnutrition among hospitalized patients with heart failure in the public and private health systems in Brazil, from April 2018 to January 2021.

Variables	Total	Private	Public	*p* ^3^
	*n* (%)	*n* (%)	*n* (%)	
**BMI** ^1^	215	67	148	0.119
Nourished	180 (83.7)	60 (89.6)	120 (81.1)	
Malnutrition	35 (16.3)	7 (10.4)	28 (18.9)	
**SGA/MNA** ^2^	247	68	179	0.793
Nourished	61 (24.7)	16 (23.5)	45 (25.1)	
Risk/Malnutrition	186 (75.3)	52 (76.5)	134 (74.9)	

Legend: ^1^ BMI: body mass index (Kg/m²). All patients classified in this way according to the BMI cut-off points adopted by the Ministry of Health were considered malnourished [28,29,30]. The following were considered nourished: other patients (eutrophic, overweight or obese); ^2^ SGA: subjective global assessment, and MNA: mini nutritional assessment; ^3^ Pearson’s chi-square and Fisher’s exact tests were performed (*p* < 0.05).

**Table 3 ijerph-19-10090-t003:** Comparison of in-hospital and post-discharge clinical outcomes (≤30 days) among hospitalized patients with heart failure, in the public and private health systems in Brazil, from April 2018 to January 2021.

Outcomes	Total	Private	Public	*p* ^2^
	*n* (%)	*n* (%)	*n* (%)	
**Hospitalization period (days)**	247	68	179	0.609
≤7	59 (23.9)	19 (27.9)	40 (22.3)	
7 to 14	80 (32.4)	22 (32.4)	58 (32.4)	
>14	108 (43.7)	27 (39.7)	81 (45.3)	
**Of treatment** ^1^	246	68	178	<0.001
Hospital discharge	191 (77.6)	65 (95.6)	126 (70.8)	
Transfer	39 (15.9)	0 (0)	39 (21.9)	
Death	16 (6.5)	3 (4.4)	13 (7.3)	
**After discharge (<30 days)**	158	47	111	0.039
None	109 (69.0)	37 (78.7)	72 (64.9)	
Readmission/Emergency	36 (22.8)	10 (21.3)	26 (23.4)	
Death	13 (8.2)	0 (0)	13 (11.7)	

Legend: ^1^ The case of evasion was not considered, that is, sample = 246 (private = 68 and public = 178). Patients were transferred for surgical procedures; ^2^ Pearson’s chi-square and Fisher’s exact tests were performed (*p* < 0,05).

**Table 4 ijerph-19-10090-t004:** Comparison of in-hospital and post-discharge (≤30 days) clinical outcomes of the sample, according to the nutritional status of hospitalized patients with heart failure, in the public and private health systems in Brazil, from April 2018 to January 2021.

Outcomes	Total	Malnutrition ^1^	Nourished	*p* ^3^
	*n* (%)	*n* (%)	*n* (%)	
**Hospitalization period (days)**	247	186	61	0.020
≤7	59 (23.9)	38 (20.4)	21 (34.4)	
7 to14	80 (32.4)	58 (31.2)	22 (36.1)	
>14	108 (43.7)	90 (48.4)	18 (29.5)	
**Of treatment** ^2^	246	185	61	0.756
Hospital discharge	191 (77.6)	144 (77.8)	47 (77.0)	
Transfer	39 (15.9)	28 (15.1)	11 (18.0)	
Death	16 (6.5)	13 (7.0)	3 (4.9)	
**After discharge (<30 days)**	158	116	42	0.634
None	109 (69.0)	79 (68.1)	30 (71.4)	
Readmission/Emergency	36 (22.8)	26 (22.4)	10 (23.8)	
Death	13 (8.2)	11 (9.5)	2 (4.8)	

Legend: ^1^ Malnutrition: all volunteers classified as at risk of malnutrition or malnutrition, according to subjective global assessment (SGA) or mini nutritional assessment (MNA); ^2^ The case of evasion was not considered. Volunteers were transferred to perform surgical procedures; ^3^ Pearson’s chi-square and Fisher’s exact tests were performed (*p* < 0.05).

**Table 5 ijerph-19-10090-t005:** Comparison of in-hospital and post-discharge (≤30 days) clinical outcomes of the sample, according to nutritional status and type of care, of hospitalized patients with heart failure, in the public and private health systems in Brazil, in the period April 2018 to January 2021.

Outcomes	Malnutrition ^1^		*p* ^3^
	Private	Public	
	*n* (%)	*n* (%)	
**Hospitalization period (days)**	52	134	0.820
≤7	10 (19.2)	28 (20.9)	
7 to 14	18 (34.6)	40 (29.9)	
>14	24 (46.2)	66 (49.3)	
**Of treatment** ^2^	52	133	
Hospital discharge	49 (94.2)	95 (71.4)	0.001
Transfer	0 (0)	28 (21.1)	
Death	3 (5.8)	10 (7.5)	
**After discharge (<30 days)**	35	81	0.071
None	26 (74.3)	53 (65.4)	
Readmission/Emergency	9 (25.7)	17 (21.0)	
Death	0 (0)	11 (13.6)	

Legend: ^1^ Malnutrition: all volunteers classified as at risk of malnutrition or malnutrition, according to subjective global assessment (SGA) or mini nutritional assessment (MNA); ^2^ The case of evasion was not considered. Volunteers were transferred to perform surgical procedures; ^3^ Pearson’s chi-square and Fisher’s exact tests were performed (*p* < 0.05).

## Data Availability

The data that support this study can be obtained from the address: www.ufs.br/Department of Physical Education, accessed on 12 June 2022.

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
