# Peer review of "Assessment of Malnutrition in Heart Failure and Its Relationship with Clinical Problems in Brazilian Health Services"

_ijerph, 2022, doi:10.3390/ijerph191610090_

Round 1

Reviewer 1 Report

The authors decided to investigate a very important clinical problem i.e. malnutrition among HF patients in connection to the type of healthcare system (public or private). The idea of the study is very good. The description of the methods is mostly fairly clear, however it needs more explanation in few aspects. The presented short-term results are interesting.

Nonetheless, I would like to add few comments:

Line 58 – The proper term for the disease is “heart failure” (without “syndrome”) as the HF is the complex disorder that leads to many symptoms. 

Line 72 – Please change the term “syndrome” to “disease”.

Line 108 – Please add the criteria used for HF diagnosis. Was it ICD code or specific parameters?

Line 113 – The sentence seems unfinished. Please correct it.

Line 222-223 – Please describe in details how the “weight of the edema” value was calculated.

Please provide relevant pages for the references 28. and 31.  

Lines 384-387 Sze et al. used different tools for nutritional status assessment and they assessed the mortality after hospital discharge for way longer time (years). Do Authors consider that this could be the reason for the inconsistence between the results from the conducted study and the study by Sze et al.?

As the 30-day follow up period could interfere the results, the longer observation would be helpful. I suggest the Authors to include this information in the limitations of the study paragraph. 

The Authors clearly pointed other limitations of their study, as there are other more adequate methods for nutritional status assessment, that were not used in this study. I suggest to point them.

Overall merit of the manuscript is good; however, the Authors should clarify their methodology and its impact on the presented results. 

Author Response

Dear Reviewer.
We hope all is well. We appreciate the considerations and made all the adjustments noted. We remain at your disposal for any other queries.

Best Regards

Reviewer 1

The authors decided to investigate a very important clinical problem i.e. malnutrition among HF patients in connection to the type of healthcare system (public or private). The idea of the study is very good. The description of the methods is mostly fairly clear, however it needs more explanation in few aspects. The presented short-term results are interesting.

Nonetheless, I would like to add few comments:

  1. Line 58 – The proper term for the disease is “heart failure” (without “syndrome”) as the HF is the complex disorder that leads to many symptoms. 

We understand your position and thank you for your remarks. The term “syndrome” was substituted by the term “disease” (Lines 72, 339, 417 e 423).

  1. Line 72 – Please change the term “syndrome” to “disease”.

We appreciate your comments. The adjustment was made (72, 339, 417 e 423).

  1. Line 108 – Please add the criteria used for HF diagnosis. Was it ICD code or specific parameters?

We thank you for your remarks. The heart failure diagnosis was obtained through the Framingham criteria. This information was included in the text (Lines 108-109).

  1. Line 113 – The sentence seems unfinished. Please correct it.

We are thankful for your observation. The adjustment was made (Line 113).

  1. Line 222-223 – Please describe in details how the “weight of the edema” value was calculated.

We appreciate your comment. The details of the “weight of the edema” discount were included in the text (Line 223-226).

  1. Please provide relevant pages for the references 28. and 31.

The detailing was included (Line 523 and 529).

  1. Lines 384-387 Sze et al. used different tools for nutritional status assessment and they assessed the mortality after hospital discharge for way longer time (years). Do Authors consider that this could be the reason for the inconsistence between the results from the conducted study and the study by Sze et al.?

We are grateful for your remarks. Yes, these differences were considered. The studies were cited because they corroborate what is commonly found in the literature. Therefore, for this aspect to become clearer, the follow-up time after hospital discharge was considered a limitation in this study (Adjusted, Line 402).

  1. As the 30-day follow up period could interfere the results, the longer observation would be helpful. I suggest the Authors to include this information in the limitations of the study paragraph.

We understand your position and thank you for your remarks. The adjustment was made (Line 402).

  1. The Authors clearly pointed other limitations of their study, as there are other more adequate methods for nutritional status assessment, that were not used in this study. I suggest to point them.

We understand your position and appreciate your comments. As explained in the article, although the body mass index (BMI) is widely used in clinical practice, both for its simplicity and for its association with cardiovascular diseases and risk of medical complications¹, its use in patients with heart failure requires care. This is because the use of BMI alone to assess nutritional status may be altered by the presence of edema, common in these patients² (Line 349-353). Thus, for this group, the use of nutritional screening tools is a viable alternative. The Subjective Global Assessment (SGA) and Mini Nutritional Assessment (MNA) were used because they are simple, low-cost and more sensitive tools for diagnosing malnutrition than BMI. In this way, they help to understand the real scenario of the nutritional status of hospitalized patients (Lines 360-361 e 405-410).

Sources:

¹Comitê Coordenador da Diretriz de Insuficiência Cardíaca. (2018). Diretriz brasileira de insuficiência cardíaca crônica e aguda. Diretriz Brasileira de Insuficiência Cardíaca Crônica e Aguda., 11(3), 436–539. https://doi.org/10.5935/abc.20180190

²Fernández-Pombo, A.; Rodríguez-Carnero, G.; Castro, A.I.; Cantón-Blanco, A.; Seoane, L.M.; Casanueva, F.F.; Crujeiras, A.B.; Martínez-Olmos, M.A. Relevance of Nutritional Assessment and Treatment to Counteract Cardiac Cachexia and Sarcopenia in Chronic Heart Failure. Clinical Nutrition 2021, 40, 5141–5155, doi:10.1016/j.clnu.2021.07.027.

Overall merit of the manuscript is good; however, the Authors should clarify their methodology and its impact on the presented results.

Reviewer 2 Report

Dear Authors, 

First of all, I would like to say that the topic is very interesting, as nutritional status is an often neglected aspect in the heart failure treatment. 

As a general recommendation, the English language should be carefully revised throughout the article, because there are omissions or expressions that are not really academic.

Specifically, I would recommend reformulation of the following lines 68-76,  78-79, 92-93 with a higher attention to English writing.

The information from lines 155-156 is not clear.

Lines 171-173 should be reformulated. The ejection fraction is indeed an important parameter, but I believe the authors intended another message and not : ,, it is associated with the presence of main etiologies''. 

From the presented results, it emerged that heart failure patients from the private sector had a better outcome than those from the public sector. This aspect may be generally valid in all countries. Perhaps it would be good to comment on the fact that patients with higher incomes who can afford private services generally have a higher functional status secondary to higher financial possibilities. This would be automatically reflected in terms of nutritional status. The question would be, is this better outcome actually the result of private services or is it associated with a better lifestyle and a higher educational level? 

Author Response

Dear Reviewer.
We hope all is well. We appreciate the considerations and made all the adjustments noted. We remain at your disposal for any other queries.

Best Regards

Reviewer 2

First of all, I would like to say that the topic is very interesting, as nutritional status is an often neglected aspect in the heart failure treatment.

  1. As a general recommendation, the English language should be carefully revised throughout the article, because there are omissions or expressions that are not really academic. Specifically, I would recommend reformulation of the following lines 68-76, 78-79, 92-93 with a higher attention to English writing.

We understand your position and appreciate your comments. All lines indicated were reviewed and adjusted.

  1. The information from lines 155-156 is not clear.

We thank you for the remark. The adjustment was made.

  1. Lines 171-173 should be reformulated. The ejection fraction is indeed an important parameter, but I believe the authors intended another message and not: ,, it is associated with the presence of main etiologies''.

We are thankful for your remark. The sentence was revised and adjusted.

  1. From the presented results, it emerged that heart failure patients from the private sector had a better outcome than those from the public sector. This aspect may be generally valid in all countries. Perhaps it would be good to comment on the fact that patients with higher incomes who can afford private services generally have a higher functional status secondary to higher financial possibilities. This would be automatically reflected in terms of nutritional status. The question would be, is this better outcome actually the result of private services or is it associated with a better lifestyle and a higher educational level? 

We appreciate the comment. For the analysis between the public and private health systems, only the characteristics of those assisted were evaluated. In this way, we have no way of properly analyzing the quality of the service. However, it is important to emphasize that the type of care was not associated with the frequency of malnutrition, but rather with treatment outcomes (transfer and death) and after discharge (emergency/rehospitalization and death). These differences observed may be a reflection of the difference in the socioeconomic profile of the public and private health networks, as explained (line 381-386) when performing the analogy with the Municipal Human Development Index (MHDI). According to Santos et al (2021), the MHDI is an adaptation of the country's global Human Development Index for the municipal and state levels, but keeping the same interpretation, that is, it is about socioeconomic development. To make this information clearer in the article, adjustments were made (lines 384-385 and 387-392).

Source: Santos, S.C.; Villela, P.B.; Oliveira, G.M.M. de Mortalidade por Insuficiência Cardíaca e Desenvolvimento Socioeconômico no Brasil, 1980 a 2018. Arq Bras Cardiol 2021, doi:10.36660/abc.20200902

Round 2

Reviewer 1 Report

Dear Authors, thank you very much for the careful revision of your manuscript. I believe that implementing suggested corrections significantly improved the quality of the paper. I am satisfied with your reply to my comments and the adjustments made in the manuscript.

I have only one more comment - line 405: I suggest adding information that the study also did not include bioelectrical impedance analysis (which is an objective recognized method for nutritional status assessment in HF patients). I understand that it was not possible to use this tool in this retrospective model study. None the less, I suggest to underline it in the limitations of the study section.

I also suggest careful revision of typos and general English.

Author Response

Dear Reviewer
We appreciate the input and made the adjustments as required. We are at your disposal for any other questions and thank you in advance.

Best Regrads

Reviewer 2

Dear Authors, thank you very much for the careful revision of your manuscript. I believe that implementing suggested corrections significantly improved the quality of the paper. I am satisfied with your reply to my comments and the adjustments made in the manuscript.

  1. I have only one more comment - line 405: I suggest adding information that the study also did not include bioelectrical impedance analysis (which is an objective recognized method for nutritional status assessment in HF patients). I understand that it was not possible to use this tool in this retrospective model study. None the less, I suggest to underline it in the limitations of the study section.

We understand your position and appreciate your considerations. Although bioelectric impedance (BIA) is a safe option to estimate body composition, its use is limited in patients with edema, common among those diagnosed with heart failure (HF).

Fernández-Pombo, Antía, Gemma Rodríguez-Carnero, Ana I. Castro, Ana Cantón-Blanco, Luisa M. Seoane, Felipe F. Casanueva, Ana B. Crujeiras, e Miguel A. Martínez-Olmos. “Relevance of Nutritional Assessment and Treatment to Counteract Cardiac Cachexia and Sarcopenia in Chronic Heart Failure”. Clinical Nutrition 40, no 9 (2021): 5141–55. https://doi.org/10.1016/j.clnu.2021.07.027.

  1. I also suggest careful revision of typos and general English.

We appreciate the observation. Adjustments were made.

This manuscript is a resubmission of an earlier submission. The following is a list of the peer review reports and author responses from that submission.